# Comparison of Nutritional Profiles of Super Worm (*Zophobas morio*) and Yellow Mealworm (*Tenebrio molitor*) as Alternative Feeds Used in Animal Husbandry: Is Super Worm Superior?

**DOI:** 10.3390/ani12101277

**Published:** 2022-05-17

**Authors:** Danka Dragojlović, Olivera Đuragić, Lato Pezo, Ljiljana Popović, Slađana Rakita, Zorica Tomičić, Nedeljka Spasevski

**Affiliations:** 1Institute of Food Technology, University of Novi Sad, Bulevar Cara Lazara 1, 21000 Novi Sad, Serbia; olivera.djuragic@fins.uns.ac.rs (O.Đ.); sladjana.rakita@fins.uns.ac.rs (S.R.); zorica.tomicic@fins.uns.ac.rs (Z.T.); nedeljka.spasevski@fins.uns.ac.rs (N.S.); 2Faculty of Technology, University of Novi Sad, Bulevar Cara Lazara 1, 21000 Novi Sad, Serbia; ljiljana04@tf.uns.ac.rs; 3Institute of General and Physical Chemistry, University of Belgrade, Studentski Trg 12-16, 11000 Belgrade, Serbia; latopezo@yahoo.co.uk

**Keywords:** *Zophobas morio*, *Tenebrio molitor*, alternative feeds, rearing optimization, full-fat meal

## Abstract

**Simple Summary:**

Currently, the food industry is facing numerous problems related to the increase in the global human population resulting in an increase in the demand for livestock. Animal feed production, as the chain leader of food production, needs to reduce the utilization of commonly used feeds, such as soybean meal and fish meal, and replace them with more sustainable ones. The utilization of insects as an alternative sustainable feed in the upcoming years can be one of the solutions. Optimization of rearing conditions, which includes the choice of insect species, time of harvest, and proper insect diet, is highly desirable for wider insect mass production. Along with the optimization of rearing conditions, insect producers will be able to obtain the desirable biomass and nutritive composition of insect products with the minimization of production costs. With their desirable nutritional composition, super worms could be used in extended mass production. Additionally, in animal feed production, super worms and yellow mealworms can be used as a nutritional source and a promising alternative to traditional feed ingredients. However, the optimization of the rearing conditions is needed for wider use in the animal feed industry.

**Abstract:**

Edible insects are acknowledged as a valuable nutritional source and promising alternative to traditional feed ingredients, while the optimization of rearing conditions is required for their wider utilization in the animal feed industry. The main goal of this study was to compare and optimize the rearing conditions of the two species’ larvae and identify the most favorable nutritive composition of the full-fat larval meal. For that purpose, *Tenebrio molitor* (TM) and *Zophobas morio* (ZM) were reared on three different substrates and harvested after three time periods. An artificial neural network (ANN) with multi-objective optimization (MOO) was used to investigate the influence between the observed parameters as well as to optimize and determine rearing conditions. The optimization of the larval rearing conditions showed that the best nutritive composition of full-fat larval meal was obtained for ZM larvae reared on a mixture of cabbage, carrot and flaxseed and harvested after 104 days. The best nutritive composition contained 39.52% protein, 32% crude fat, 44.01% essential amino acids, 65.21 mg/100 g Ca and 651.15 mg/100 g P with a favorable ratio of 1.5 of n6/n3 fatty acids. Additionally, the incorporation of flaxseed in the larval diet resulted in an increase in C18:3n3 content in all samples.

## 1. Introduction

The current food and feed industry is facing numerous problems related to the increase in the global population, economic development and air and water pollution. The global population will reach more than 10 billion people by 2050, and the demand for animal protein will continue to increase in the following years [1]. As the leader of food manufacturing, animal feed production needs to consider and include alternative and sustainable feed sources. In the past eight years, the price of protein sources as animal feed ingredients has doubled and already represents 60–70% of the total costs of animal feed production [2]. Ever since 1975, when Meyer-Rochow suggested the use of edible insects as food and feed that could ease the problem of global food shortages, insects have become recognized as an attractive nutritional source and in connection with animal husbandry as well [3]. This is mainly due to the insects’ fast growth in small enclosures, high feed conversion and lower emissions of greenhouse gases and ammonia [4]. What is more, insects do not compete with food resources, making their production highly sustainable and cost-effective [5]. Additionally, some edible insects, such as *Allomyrina dichotoma*, *Tenebrio molitor*, *Protaetia brevitarsis*, *Gryllus bimaculatus*, *Teleogryllus emma* and *Apis mellifera*, have shown beneficial biological properties in human health [6]. However, insect mass production is still very challenging. In order to become competitive with commonly used feed ingredients (i.e., soybean meal and fishmeal), and to achieve sustainability in the animal feed industry, expanded mass production is highly preferable [7]. There are only a few companies in the world that claim to have a capacity higher than 10 t of insect meal per day, yet mass production of insects is still increasing [8]. That being the case, it is necessary to provide automated insect rearing facilities with an optimized production process. This will increase the scale of insect production, followed by continuous quantity and quality, and reduce the cost of insect production and processing [9,10]. Insects represent natural food for various species, including poultry, pig and fish. Moreover, Commission Regulation (EU) 2017/893 has authorized the use of processed animal protein derived from insects and compound feed containing such processed animal protein in fish, pets, poultry and pig diets, which encourages wider insect mass production [11].

Besides *Hermetia illucens*, *Tenebrio molitor* (TM) is one of the most examined insects in animal feed science [12]. The main advantages of wider mass production of TM, also known as yellow mealworm, are that females lay an abundance of eggs (approximately 500), the life cycle from egg to adult is relatively short and feed requirements are not demanding [13]. Many studies have already observed the influence of a TM-enriched diet on growth performances, weight gain and feed efficiency of broilers, laying hens and fish, where it was found that with proper diet composition, TM could be used as an alternative protein source [9,10,14]. One of the more recently investigated insects with great nutritional potential for future use as an alternative feedstuff is *Zophobas morio* (ZM), also known as super worm [15,16]. Female ZM lay more than 2000 eggs, and rearing conditions are very similar to TM, but reliable scientific data regarding its mass production are still largely lacking [16]. Likewise, to the authors’ best knowledge, published data on the nutritive evaluation of ZM and its integration into the diet of various animal species is still limited. In some studies, ZM has been used as a feedstuff in broiler, fish and pig diets and it has been shown that it can have a positive influence on their growth performances, feed intake and amino acid (AA) profile [14,16,17]. TM and ZM contain a high percent of protein and lipids and can be considered an abundant source of essential AA, polyunsaturated fatty acids (PUFA), minerals, fibers, bioactive compounds and carotenoids [18,19]. Additionally, chitin and some peptides obtained from TM and ZM have shown antimicrobial and prebiotic properties, which positively influence animal health [20,21]. It was also reported that a low inclusion level of TM and ZM full-fat meals improved the growth performance and immune system traits of broiler chickens [22].

Numerous food-related investigations introducing the effectiveness of ANN modeling could be found in the literature by Cetin and Saglam [23] and Turkoglu et al. [24]. Moreover, in a study by Wenning et al. [25], automatic monitoring of the insects’ growth system was developed using a mask scoring regional convolutional neural network. Using the obtained neural networks makes it possible to measure house cricket (*Acheta domesticus*) characteristics, such as shape and color, and it is possible to predict their age, size and health. In a study by Baur et al. [26], an automated system for the breeding of TM was developed for the purpose of protein production. In this investigation, a combination of classical image processing functions and a neural network was presented. The output variable was harvesting time. In a book chapter by Kröncke et al. [27], the automation system of insect mass rearing and processing technologies of mealworms (TM) was developed according to the ANN model. The results of the ANN model for the automated separation and monitoring of the breeding process of mealworms have shown that it is possible to separate mealworm larvae from the substrate and feces effectively with a zig-zag air separator and make handling and separating processes easier.

Some papers indicate that the diet, insect species and processing method have a notable impact on the nutritional value of the various insect meals [28,29]. In order to obtain a nutritional value of insect meal that would meet animals’ requirements and potentially replace traditional feed ingredients, it is of the highest interest to optimize the insect diets [30,31]. Therefore, the objective of this work was to evaluate the possibility of predicting a nutritive profile in terms of proximate composition, AA content, mineral content and fatty acid (FA) profile of full-fat larval meal as a function of the insect species (TM or ZM), the larval diet and time of larvae harvest. Moreover, the multi-objective optimization (MOO) was employed for adjusting the rearing conditions that will result in the optimization of the nutritional composition of the full-fat larval meal, along with the use of an artificial neural model (ANN) and genetic algorithm (GA), knowing that there could not be a single solution due to conflicting objective functions [32]. 

## 2. Materials and Methods

### 2.1. Materials

Super worm (*Zophobas morio*, Coleoptera: Tenebrionidae) and yellow mealworm (*Tenebrio molitor*, Coleoptera: Tenebrionidae) were selected for this study. TM and ZM were reared only on a plant-based diet, which is in accordance with the EU legislation where the inclusion of animal-based products in an insect’s diet, which will further be used in animal nutrition, is not allowed [33]. Therefore, three different formulations of diet were used in the experiment. Each diet included wheat bran (850 g) and a variation of carrot, cabbage or a mixture of carrot, cabbage and flaxseed. Carrots and cabbage were collected as waste from a local restaurant, while wheat bran and flaxseed were purchased from a local market. Subsequently, flaxseed was milled, and vegetables were chopped into smaller pieces.

### 2.2. Rearing Conditions 

Adult beetles were allowed to deposit eggs for three weeks. Afterward, identical amounts of larvae (42.60 g) were collected and placed in plastic boxes, which were previously filled with wheat bran. Larvae were reared under laboratory-scale production in plastic boxes, dimension 28 × 40 × 13.5 cm. Larvae were fed every third day with 100 g of carrot (100%), cabbage (100%) or a mix of carrot, cabbage and flaxseed (1:1:1). Spoiled vegetables were replaced with fresh ones, and dead larvae were removed. The temperature in the room remained at 27 ± 2 °C with 50–60% of humidity. Subsequently, larvae were reared and harvested for 90, 97 and 104 days. Larvae were separated from the substrate, beetles and pupae with different sieve types (4 and 5.613 mm) and, afterward, starved for 24 h in order to separate them from feces. Then, to manage microbiological quality, the larvae were inactivated in boiling water for 30 s, dried for 1 h at 70 °C and milled in a water-cooled laboratory mill (KN 295 Knifetec^™^, FOSS, Hilleroed, Denmark). The inactivating method was chosen on the basis of a recommendation by Singh et al. [34] and Larouche et al. [35]. 

### 2.3. Proximate Analysis

Protein content, crude fat (CF), crude ash and phosphorous were determined according to the standard AOAC methods [36], while protein digestibility (further digestibility) was analyzed by the AOAC method [37]. Crude fiber was determined using the fiber analyzer (Ankom 220, ANKOM Technology, New York, NY, USA). Nitrogen-to-protein conversion factor (kp) in the case of whole larvae should be 4.76 to avoid protein overestimation due to non-protein nitrogen from chitin [38]. Therefore, a kp of 4.76 was further used for protein determination. Results were expressed on a dry matter.

### 2.4. Amino Acid Analysis

AA analysis was performed on samples by ion-exchange chromatography using an automatic AA analyzer Biochrom 30+ (Biochrom, Cambridge, UK), according to Spackman et al. [39]. The technique was based on AA separation using strong cation exchange chromatography, followed by the ninhydrin color reaction and photometric detection at 570 nm and 440 nm (for proline). Samples of the full-fat larval meal were previously hydrolyzed in 6 M HCl (Merck, Darmstadt, Germany) at 110 °C for 24 h. Essential amino acid tryptophan was not determined because it was destroyed during acid hydrolysis in 6 M HCl. AA content was expressed as % of AA in protein content.

### 2.5. Mineral Analysis

Mineral content of larvae (Ca, Na, Mg, K, Fe, Zn, Mn, Cu) was determined by using an atomic absorption spectrometer (Varian Spectra AA 10, Varian Techtron Pty Limited, Mulgrave, Victoria, Australia) equipped with a flame furnace and operated with air acetylene flame. Results were expressed as mg of mineral content on 100 g of dry matter.

### 2.6. Fatty Acid Analysis

Larval oil was obtained by extraction with a 2:1 chloroform-methanol mixture. Fatty acid methyl esters (FAMEs) of samples were analyzed by gas chromatography (GC) on an Agilent 7890A system (Agilent Technologies, Santa Clara, CA, USA) with a flame ionization detector (GC-FID), auto-injection module for liquid, equipped with a fused silica capillary column (Supelco SP-2560 Capillary GC Column 100 m × 0.25 mm, d = 0.20 μm) (Supelco, Bellefonte, PA, USA) and helium as a carrier gas (purity = 99.9997 vol %, flow rate = 1.5 mL/min and pressure = 1.092 bar). The principle behind the FAMEs is the comparison between retention time and reference standards (Supelco 37 component FAME mix, Sigma Aldrich, Darmstadt, Germany). The final results were expressed as % of individual FA or FA group in total identified FA.

### 2.7. ANN Modelling

The multi-layer perceptron model (MLP), with three layers (input, hidden and output), was applied to the ANN model calculation. This form of the ANN model is wide-appreciated for its high capability of approximating nonlinear functions [40,41]. Prior to calculation, input and output data needs to be normalized (the Min-Max normalization method was used in this paper) in order to improve the conduct of the ANN [42]. During the calculation of the ANN, input data are repeatedly introduced to the network [42,43]. The training process of the network was repeated 100,000 times, testing different topologies of ANN, with a different number of neurons in hidden and output layers (5–20), with different activation functions (such as logarithmic, logistic, tangent hyperbolic or identity), and with random initial values of weight coefficients and biases. The optimization of the ANN structure was performed by minimizing the validation error. The Broyden–Fletcher–Goldfarb–Shanno algorithm was applied to find the solution to the unconstrained nonlinear optimization during the ANN modeling [42]. 

### 2.8. Global Sensitivity Analysis

The Yoon’s global sensitivity equation was used to calculate the relative impact of the input parameters on output variables according to weight coefficients of the developed ANN models [44]:(1)$$RI_{ij}\left(\%\right)=\frac{\sum_{k=0}^{n}\left(w_{ik}\cdot w_{kj}\right)}{\sum_{i=0}^{m}\left|\sum_{k=0}^{n}\left(w_{ik}\cdot w_{kj}\right)\right|}\cdot 100\%,$$
where *RI*—relative impact, *w*—weight coefficient in ANN model, *i*—input variable, *j*—output variable, *k*—hidden neuron, *n*—number of hidden neurons, *m*—number of inputs. 

### 2.9. Statistical Analyses

The data were processed statistically using the software package STATISTICA 10.0 (StatSoft Inc., Tulsa, OK, USA). All determinations were made in triplicate; all the data were averaged and expressed by mean values. 

### 2.10. Multi-Objective Optimization (MOO)

The developed ANN model was used for MOO with the goal of finding rearing conditions that will result in maximal proximate composition, AA content, mineral content and essential FA profile, but the minimal ash content and n6/n3 ratio. The solution of MOO was a Pareto front which existed only if it would improve one objective function without worsening others [32]. The genetic algorithm (GA) was used to find a solution to the MOO problem using a stochastic method inspired by natural evolution applying the mutation, selection, inheritance and crossover [45]. The MOO calculation was performed in Matlab software, using the gamultiobj function. The initial population was randomly generated and afterward presented to a set of points in the design space. The populations of the next generations were calculated by the distance measure and non-dominated ranking of the individual points in the present generation [32,45].

## 3. Results

### 3.1. ANN Models

Four ANN models (ANN1, ANN2, ANN3 and ANN4) were developed in order to predict the four groups of output variables (six proximate composition parameters, nine AA, five mineral content parameters and three FA profile parameters) based on the three input parameters (insect species, substrate and time of harvest). All input variables were categorical (non-numerical), which approved the intention of the ANN modeling approach. The obtained ANN models also were used to investigate the relative influence of 3 input variables on 23 output variables (in total), using Yoon’s equation (Equation (1)). Furthermore, the optimization of output parameters predicted in developed ANN models was conducted using the MOO approach. The MOO optimization was necessary, having in mind a huge number of parameters to be simultaneously optimized, with respect to appointed conditions that maximize the proximate composition (except ash content), AA content, mineral content and essential FA parameters, but also the minimum of ash content and n6/n3 ratio should be obtained.

ANN1 was developed to predict proximate composition variables, such as mass yield, protein content, CF, crude ash and crude fiber content and digestibility of full-fat larval meals. ANN2 was developed to predict the AA content: essential AA-threonine (Thr), valine (Val), methionine (Met), isoleucine (Ile), leucine (Leu), phenylalanine (Phe), histidine (His), lysine (Lys) and non-essential AA tyrosine (Tyr). ANN3 was developed to predict the mineral content of Ca, K, Mg, Na and P. While ANN4 was developed to predict the essential FA content of the samples: C18:2n6c (LA, linoleic acid, n6), C18:3n3 (ALA, α-linolenic acid, n3) and n6/n3 ratio.

The obtained ANN1, ANN2, ANN3 and ANN4 models demonstrated an adequate generalization capability for experimental data prediction. According to ANN1, ANN2 and ANN3 performances, the optimal number of neurons in the hidden layer for prediction of proximate composition, AA composition and mineral content were 10 (networks MLP 8-10, MLP 8-10-9, MLP 8-10-5, respectively), to obtain high values of *r*^2^ (overall 1.000 for the training period) and the low sum of squares values (SOS). Based on the ANN4 performance, it was observed that the optimal number of neurons in the hidden layer for the prediction of FA content was seven (network MLP 8-7-3) to obtain high values of *r*^2^ (1.000 for the training cycle) and low values of SOS. ANN models were used to predict experimental variables reasonably well for a broad range of values (the experimentally measured and ANN model predicted values are presented). ANN1, ANN2, ANN3 and ANN4 models are complex (156, 189, 145 and 87 weights-biases, respectively) according to the high nonlinearity of the investigated system [41].

### 3.2. Proximate Composition

The nutritive profile of TM and ZM are presented in Appendix A. Protein content was in the range of 37.27 to 42.34% in TM and in the range of 33.94 to 42.82% in ZM. ZM, the diet of which included cabbage for 97 days, showed a positive effect on the protein content. CF content varied from 24.83 to 33.54% in TM and 32.35 to 44.48% in ZM. In the case where TM was fed with carrots and harvested after 104 days, a negative effect on the CF content was apparent. Crude ash content in TM was higher than that of ZM and cabbage and a harvest time of 97 days negatively influenced crude ash content in TM. Crude fiber content in TM and ZM varied from 5.28 to 9.12%, and a positive effect on crude fiber content was found in both kinds of larvae, in which the diet included carrot and a harvest time of 97 days. Digestibility for TM and ZM was in the range of 83.05 to 85.59% and 82.05 to 84.05%, respectively. Similar to the protein content, a positive effect was found on digestibility for ZM, the diet of which included cabbage for 90 days. 

The mass yield was higher for TM compared to ZM and varied from 178.41 to 326.68 g. However, the mass yield was more uniform for ZM and was between 111.88 and 177.03 g. According to Figure 1, the yield parameter was better for TM than ZM, while the positive impact of time was found when TM larvae were harvested after 104 days of rearing.

### 3.3. Amino Acid Composition

AA profile of TM and ZM is shown in Appendix A. Regardless of the type of diet and time of harvest, Leu, Lys and Val were the most abundant essential AAs in both TM and ZM, while Met was the least abundant. The Glu, Tyr, Pro and Asp were the most dominant among non-essential AAs in both larval meals. As shown in Figure 2, the larval diet that included carrots positively affected Thr, Met, Lys and Tyr content in TM, while negatively affecting Val, Leu, Isl and His content in TM. In ZM, a positive effect was noticed in connection with Leu, His, Val and Ile content in ZM, following the use of cabbage in the larval diet during the 97 days of rearing. On the other hand, a negative effect on the Met and Thr content was apparent in ZM while using cabbage and a mixture of cabbage, carrot and flaxseed in the diet, respectively, for 97 days. The positive effect on Phe content was found when using cabbage in the TM diet for 97 days.

### 3.4. Mineral Content

The mineral content of TM and ZM is demonstrated in Appendix A. Compared with TM, ZM had a higher amount of Ca, but lower amounts of K, Mg and Zn. The Ca content for TM and ZM ranged from 24.30 to 32.28 mg/100 g and 38.68 to 66.85 mg/100 g, respectively. Cabbage positively influenced Ca content in ZM during the 104-day harvest. K content for TM ranged from 741.74 to 878.14 mg/100 g, while that for ZM ranged from 493.19 to 665.71 mg/100 g. As shown in Figure 3, cabbage and 104 days of rearing positively affected K content in both observed larvae. Mg content for TM ranged from 181.19 to 277.14 mg/100 g, while in ZM ranged from 71.95 to 100.98 mg/100 g. A negative influence on Mg content was found when ZM was fed a mixture of cabbage, carrot and flaxseed for 90 days. TM had a higher level of P than ZM, and it ranged from 751.56 to 915.80 mg/100 g for TM and 541.14 to 651.15 mg/100 g for ZM. A positive effect on P content was noticed in TM, whose diet included a mixture of cabbage, carrot and flaxseed for 104 days of rearing. TM and ZM had a similar amount of Na, Fe, Mn and Cu. In addition, a diet that included a mixture of cabbage, carrot and flaxseed had a positive effect on Na and P contents in TM for 97 days of rearing (Figure 3). 

### 3.5. Fatty Acid Composition

The FA profiles of TM and ZM are shown in Appendix A. Regardless of the insect diet and time of harvest, ZM had a higher content of saturated fatty acids (SFA) and a lower content of monounsaturated fatty acids (MUFA) than TM. SFA content for TM ranged from 21.8 to 26.3%, while that for ZM ranged from 34.9 to 43.4%. MUFA content for TM varied between 41.0 and 49.6%, whereas that for ZM was between 30.9 and 34.2%. PUFA content for TM and ZM ranged between 24.1 and 36.9%. Regardless of the insect species and harvest time, the highest content of PUFA was observed when larvae were fed with a mixture of cabbage, carrot and flaxseed. Likewise, the content of C18:3n3 was highest when the larvae had consumed a mixture of cabbage, carrot and flaxseed. The n6/n3 ratio for TM varied between 1.6 and 26.9, whereas that for ZM was in the range of 1.4 to 21.4%. A positive effect on C18:2n6c content in TM was observed when the diet included cabbage and carrots during 97 and 104 days of rearing (Figure 4). Additionally, rearing conditions, which included a mixture of cabbage, carrot and flaxseed and a time of 90 days, had a positive effect on C18:3n3 content in ZM (Figure 4). 

### 3.6. Multi-Objective Optimization of the Rearing Conditions of the ANN

The calculated optimal values of output variables (proximate composition, AA composition, mineral content and FA profile) are presented in Appendix A. Results obtained by using MOO showed that the optimal larval meal as a valuable nutritional feedstuff was ZM, reared on a mixture of cabbage, carrot and flaxseed and harvested after 104 days. These results indicate that larval meal ZM is superior to feedstuff than TM.

## 4. Discussion

### 4.1. Proximate Composition

In this study, TM exhibited a higher mass yield than ZM, and it increased with the time of rearing. Mass yield for ZM was not affected by the type of the diet and time of rearing, and it remained relatively constant. Contrary to our results, Harsányi et al. [30] reported that the mass of ZM reared on low-value waste increased more than TM during the 45-day experiment period. This discrepancy in the results could be attributed to the fact that TM larvae have an intensive growth from 77 to 125 days [46]. Additionally, enriched wheat bran with carrot and cabbage in an insect diet can improve larval weight by more than 40% [47]. 

Generally, both TM and ZM are nutritionally rich in proteins. TM and ZM had similar protein content, while the best optimization was reached for ZM fed with dietary supplementation with cabbage within 97 days. Cabbage can provide valuable micronutrients as it is nutritionally rich in ascorbic acid, a vitamin that improves an insect’s growth and fertility by reducing oxidative stress. On the other hand, it is deficient in macro-nutrients, such as proteins and lipids, and therefore, it probably does not influence the protein content in larvae [47]. However, our results showed that the interaction of diet and time of harvest could improve protein content in the larval meal. The obtained results of protein content were in agreement with or higher than the results reported by others [16,31,46]. Larvae fed with wheat bran have a higher protein content than CF content due to the higher fiber content of wheat bran, which limits the nutrition and leads to the usage of stored larval oil for their normal development processes [48]. In comparison to commonly used protein feed ingredients, such as soybean meal and fishmeal, the investigated larval meals can be considered a valuable source of protein in animal nutrition [49,50]. However, both observed larvae had less protein content compared with soybean meal (49–56%) and fishmeal (60%) [50]. ZM and TM contain high amounts of fat, and although mostly observed as a protein source, they could be considered oil sources as well. The results of CF are in agreement with previously published results [19,51,52] but lower than those reported by Harsányi et al. [30], who reared larvae on vegetable or green garden waste with grass (43.20–45.20%). Ravzanaadii et al. [53] reported CF content similar to our result (32.7%) when larvae of TM were fed wheat bran, cabbage, reddish and carrot. Both larval meals have higher CF content compared to soybean meal (3–5%) and fishmeal (10 %) [52]. The variations in the protein and CF content were low in spite of the changes in the type of substrate used. Van Broekhoven et al. [54] observed no significant variations in larval protein content despite that the level of protein content in the diets differed 2–3 fold, but reported a significant influence of the dietary fat on the larval CF content. Oonincx et al. [29] reported that the basic chemical composition (proteins, CF and moisture) of TM was not affected by cereal-based diets, whereas a diet enriched with unsaturated FA altered the FA profile. Additionally, the inclusion of carrot in some insect diets, such as *Gryllus assimilis*, did not affect the protein, lipid and ash content, but, on the contrary, it affected the FA composition [55]. Harsányi et al. [30] reared insects on different organic waste and ranked garden waste as the average feed for insect larvae, vegetable waste as the poorest, and chicken feed was marked as the best. It was observed that different insect species differ in macronutrient composition when reared on different substrates, indicating that substrates of low nutrient value can influence decreased protein content and elevated fat content in observed species [28]. Vegetables are highly used as a feed for TM as a source of water, vitamins, essential FA and sterols [10]. Recently, a high percentage of mixed vegetable waste, garden waste or cattle and horse manure were not observed as optimal rearing substrates for rearing TM and ZM larvae [30]. 

The crude ash content was higher and more variable in TM than that in ZM. The highest amount of ash content observed in this study was still lower than that of soybean meal and fishmeal [10]. Under these conditions, the lowest ash content was obtained in ZM, which is preferable in feed formulation. The content of crude ash in animal feed materials is limited by EU Regulation EC No 767/2009 [56].

Both larvae had crude fiber content similar to that of soybean meal and higher than that of fishmeal [52]. A higher amount of crude fiber has a favorable influence on the gut microbiota of animals since it is mainly composed of chitin [57]. Chitin represents the main fiber in insects, and although it is degradable only to some extent for most animals, it has beneficial prebiotic properties on animal health [58]. Recent studies have shown the potential antimicrobial, antifungal and antiviral activity of chitin and chitin derivates [59]. Chitin also showed a positive effect on the immune system of fish and poultry [9]. 

Digestibility was similar in both larvae and was mostly unaffected by the change in the diet and time period. However, similar to protein content, the obtained results showed that the supplementation of cabbage for 97 days could improve protein digestibility in ZM. Protein digestibility in TM and ZM was similar to that of soybean meal and a bit lower than that of fishmeal [58,60]. The obtained digestibility values were higher than those for dried and milled larvae of the yellow mealworm (65.5–66.7%) [61] while slightly lower than those in freeze-dried yellow mealworms and super worm (91.3 and 93.0%, respectively) [62]. It was established that insect processing and drying methods (temperature and time dependence) are factors that can have a substantial effect on protein digestibility values [29].

### 4.2. Amino Acid Composition

TM and ZM represent a valuable protein source in the animal diet as they are characterized by their higher amounts and quality of proteins and AA, comparable to some extent to traditional feed ingredients. Protein quantity and quality are some of the most important criteria to be considered in the selection of protein sources in animal diets [63]. One of the criteria that determine the quality of protein is the AA composition. Previous studies are in agreement with our results, which declare that the AA pattern is uniform for insect species and is not influenced by the diet or life stage [64]. In comparison to soybean meal, both observed larvae had preferable AA profiles, similar to those of fish and poultry meal [13]. In comparison with fishmeal, larvae of ZM and TM contain lower amounts of Met and Lys, and for further use as feedstuff, this could be a limiting factor due to their low concentrations [13]. However, through genetic methods, such as selection and hybridization of insects, it could be feasible to enhance their content in larval meal [14]. The AA composition of analyzed larval meals was in agreement with that reported by other authors [16,51]. The content of Lys and Met was slightly higher than those reported by Jabir et al. [65], but lower than those reported by Rumpold and Schlüter [18]. Prachom et al. [16] reported that the profile of essential AA of ZM was sufficient to meet known requirements for most marine species. 

### 4.3. Mineral Content

The content of Ca was higher and more variable in ZM compared with that of TM, whereas TM had higher contents of K, Mg, Zn and P than ZM. Data on the mineral composition of the larval meals are in compliance with other published results [10,14]. Siemianowska et al. [13] selected oat flakes and vegetables (not specified) as nutritional substrates for larvae of TM and reported similar contents of K, Ca and P, but lower contents of Na and Mg in powdered larval meal than those in our study. A lower content of Ca and Mg was also observed by Araújo et al. [66] when ZM was reared on wheat, corn, soybean meal, fruits and vegetables. These differences in mineral profile might be attributed to differences in nutrient composition of the available larval diet. In general, insects can be observed as feed ingredients with poor content of Ca and K, but high in P content [14]. In comparison to black soldier fly, ZM and TM contain a smaller amount of Ca content [14]. Low Ca content is an issue for further use of larval meal as feedstuff since Ca deficiency can cause symptomatic bone diseases in animals [13]. One of the solutions to meet animals’ requirements of Ca could be the utilization of a Ca-fortified diet in the insect diet. Anderson [67] reported that the application of CaCO_3_ in the insect diet could increase the Ca content in insects to desirable levels.

### 4.4. Fatty Acid Composition

ZM had a higher content of SFA but a lower level of MUFA than TM. C18:2n6 was a dominant PUFA in both larval meals, while C18:3n3 was also presented in non-negligible amounts. Likewise, FA profiles of TM and ZM followed the variations in the dietary FA content. Namely, dietary inclusion of flaxseed in insect diet strongly increased the level of C18:3n3 (more than 10-fold) and decreased the ratio of n6/n3 approximately to 2. Dietary inclusion of PUFA and, in particular, n-3 fatty acids have many beneficial effects on animal and human health, such as the prevention of cardiovascular diseases, arthritis and diabetes [68]. The ratio of n6/n3 FA should be lower, and the ratio less than 4 is considered optimal for human health [58,68]. TM and ZM are rich in essential FA, and by manipulating the feed, it is possible to increase the C18:3n3 content and decrease the n6/n3 ratio to a desirable level [64]. A more favorable level of C18:3n3 and n6/n3 ratio was found in our study than those reported by Sánchez-Muros et al. [9] and Boulos et al. [51]. The obtained results indicated that a diet abundant in C18:3n3 (i.e., flaxseed) could augment C18:3n3 content in larval meals, which agrees with earlier studies [10]. Moreover, St-Hilaire and Cranfill [69] found that by supplementing diets for black soldier fly with ingredients rich in C18:3n3, such as fish offal was a way to modify and enhance a final insect product. With the incorporation of insect meals as a source of fats and FA in animal diet, it is feasible to meet the requirements in essential FA, which can further affect the quality and health properties of animal-based products (meat and eggs).

## 5. Conclusions

Full-fat larval meals obtained from *Tenebrio molitor* and *Zophobas morio* are rich in protein, fat, essential amino acids, fatty acids and some minerals. The nutritive profile of larval meals was notably affected by the rearing conditions. Interaction of supplemented cabbage diet with proper harvest time can improve protein content and protein digestibility of *Zophobas morio* larval meal. Carrots had a negative effect on the fat content in *Tenebrio molitor* larval meal, while positively affecting the crude fiber in both larval meals. The supplemented diet with a mixture of cabbage, carrot and flaxseed positively affected most minerals in larval meals. The larval diet that included cabbage positively affected a wider number of essential amino acids but, at the same time, had a negative impact on the limited essential amino acid methionine. Furthermore, the incorporation of flaxseed in the larval diet resulted in an increase in C18:3n3 content (approximately 10-fold) and a decrease in the n6/n3 ratio to 2. The optimization of larvae rearing conditions revealed that the optimal nutritive profile of full-fat larval meal as an alternative feedstuff was obtained for *Zophobas morio* with a cabbage, carrot and flaxseed diet and harvest time of 104 days of rearing. This indicates that the larval diet should be enriched with cabbage, carrot and flaxseed. Comparisons between the nutritional properties and chemical compositions of *Tenebrio molitor* and *Zophobas morio* have shown that the latter is potentially more suitable for practical use in the feed formulation than the former.

## Figures and Tables

**Figure 1 animals-12-01277-f001:**
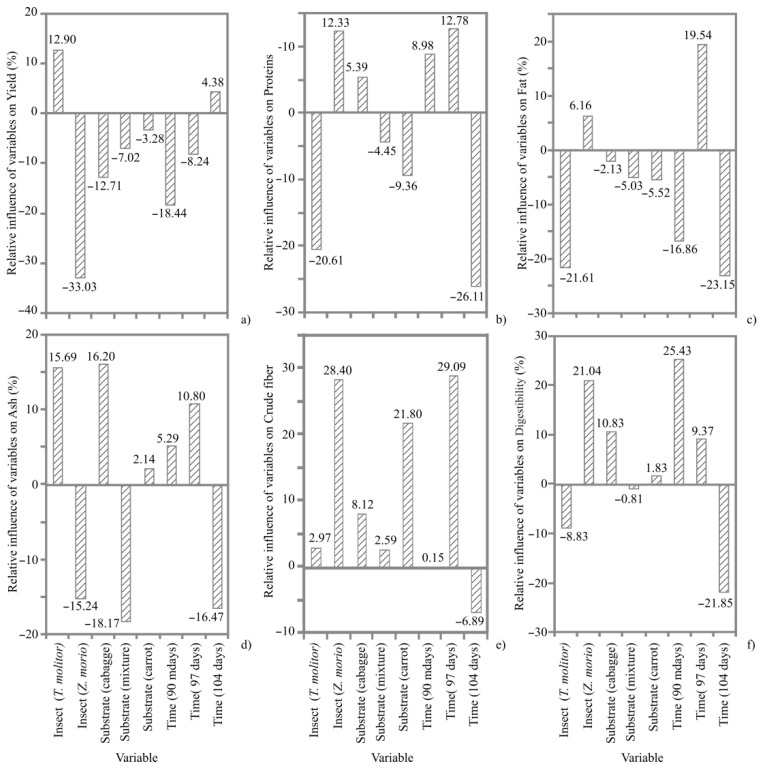
The relative influence (%) of the insect species (*Tenebrio molitor*, *Zophobas morio*), substrate (cabbage, mixture of cabbage, carrot and flaxseed and carrot) and time of harvest (90, 97 and 104 days) on proximate composition: (**a**) yield (mass yield); (**b**) protein content; (**c**) crude fat; (**d**) ash; (**e**) crude fiber; (**f**) protein digestibility determined using Yoon interpretation method.

**Figure 2 animals-12-01277-f002:**
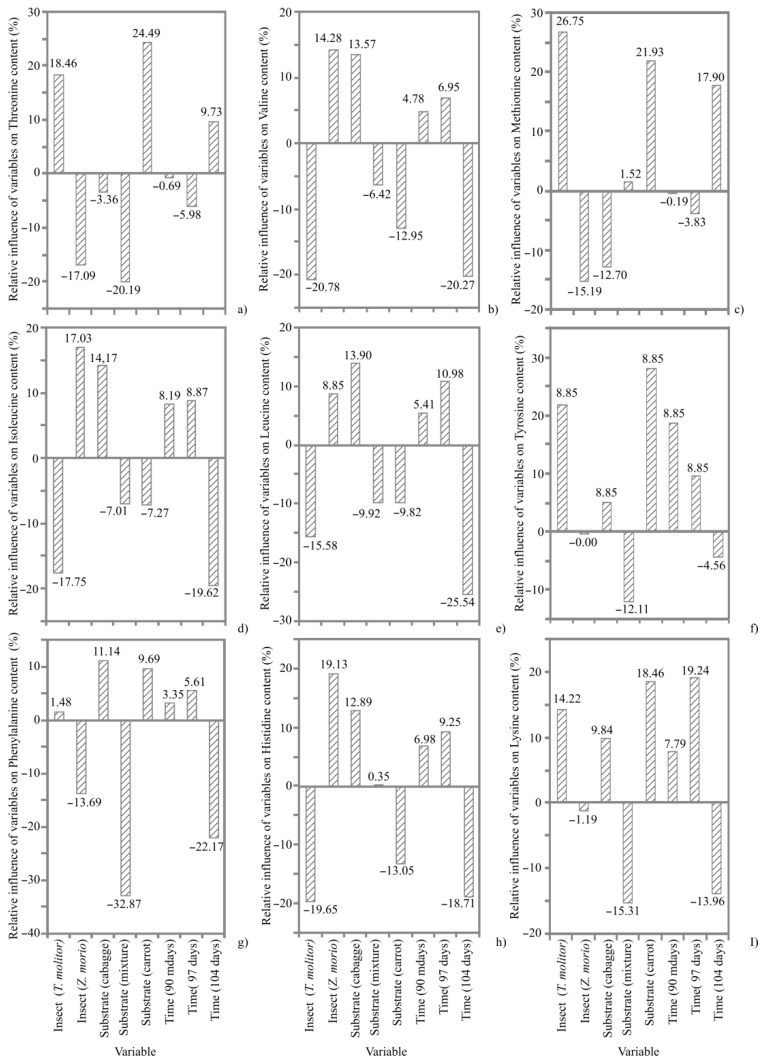
The relative influence (%) of the insect species (*Tenebrio molitor*, *Zophobas morio*), substrate (cabbage, mixture of cabbage, carrot and flaxseed and carrot) and time of harvest (90, 97 and 104 days) on the AA content: (**a**) threonine; (**b**) valine; (**c**) methionine; (**d**) isoleucine; (**e**) leucine; (**f**) tyrosine; (**g**) phenylalanine; (**h**) histidine; (**I**) lysine determined using Yoon interpretation method.

**Figure 3 animals-12-01277-f003:**
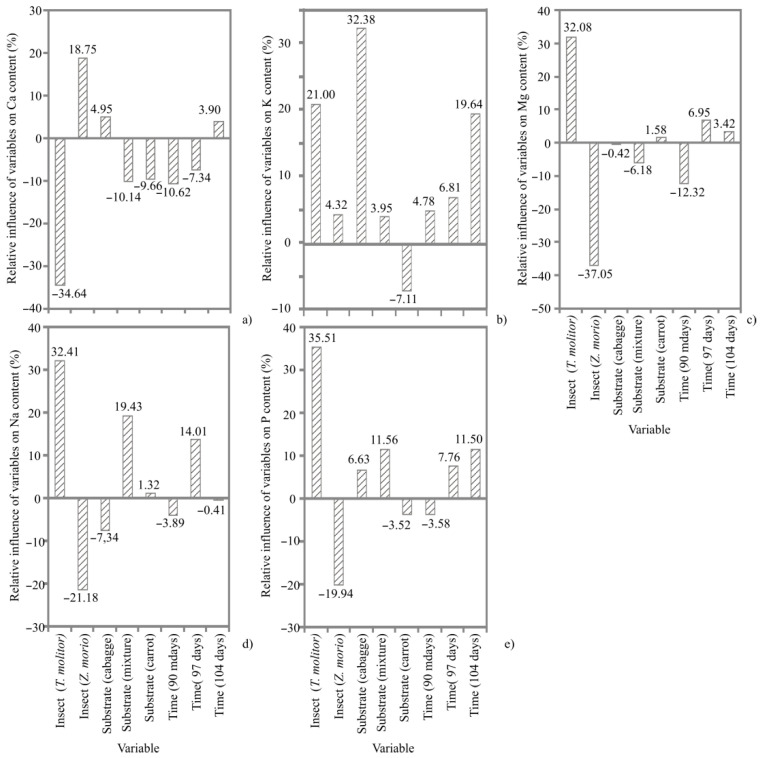
The relative influence (%) of the insect species (*Tenebrio molitor*, *Zophobas morio*), substrate (cabbage, mixture of cabbage, carrot and flaxseed and carrot) and time of harvest (90, 97 and 104 days) on the mineral content: (**a**) Ca; (**b**) K; (**c**) Mg; (**d**) Na; (**e**) P determined using Yoon interpretation method.

**Figure 4 animals-12-01277-f004:**
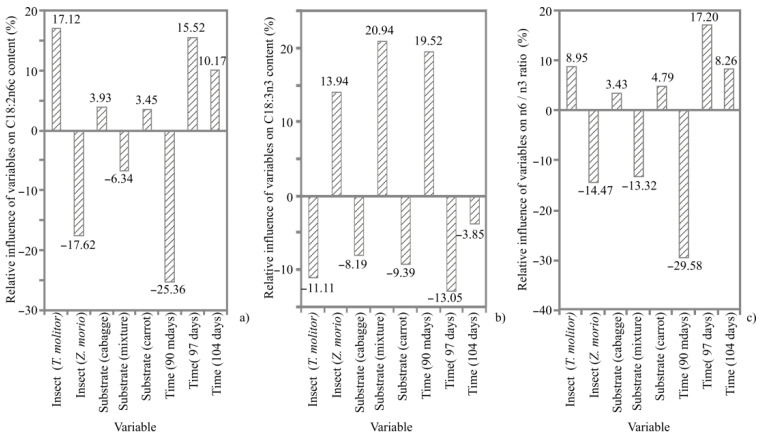
The relative influence (%) of the insect species (*Tenebrio molitor*, *Zophobas morio*), substrate (cabbage, mixture of cabbage, carrot and flaxseed and carrot) and time of harvest (90, 97 and 104 days) of the FA content: (**a**) C18:2n6c; (**b**) C18:3n3; (**c**) n6/n3 ratio determined using Yoon interpretation method.

## Data Availability

Data is contained within the article and the Appendix A.

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
