# Peer review of "Comparison of Nutritional Profiles of Super Worm (Zophobas morio) and Yellow Mealworm (Tenebrio molitor) as Alternative Feeds Used in Animal Husbandry: Is Super Worm Superior?"

_animals, 2022, doi:10.3390/ani12101277_

Round 1
Reviewer 1 Report
The manuscript was greatly improved in respect to readability but still some improvements are needed.
Lines 35-36 Use unform units here - % or g/100 g – and along the manuscript.
Lines 114-115 Do not use the designation substrate K but use cabbage. Do not use the designation substrate S but use carrot. Here and along the manuscript.
Line 213 Place the sentence “According to Figure 1, 213 yield parameter was better for TM than for ZM while the positive impact of time was 214 found when larvae TM were harvested after 104 days of rearing. “ on new paragraph at the end of this section 3.1, with na introduction for the type of results presented in Fig. 1 (what Yoon’s method delivers).
Line 389 Delete “one of the”.
Conclusions
Improve this section. What are the ideal diets? What substrate influences the composition of the insect on specific nutrient? You conclude about the flaxseed but can’t you conclude about others?
Author Response
Please see the attachment (Reviewer 1)

Reviewer 2 Report
Title: Comparison of nutritional profile of super worm (Zophobas morio) and yellow mealworm (Tenebrio molitor) as alternative feeds in contemporary animal feed production. Is SUPER WORM SUPERIOR?
Authors: Danka Dragojlović et al.
The manuscript by Danka Dragojlović et al. has potential, but at the moment it is totally unacceptable on account of its numerous flaws. It requires a major revision, before the manuscript can be considered again. Although one issue is the English (grammar, sentence construction, style) another issue concerns the contents. The objective is clear and the results are convincing, although this reviewer wonders why vitamins (measured as important components in the diets of the larval mealworms on Line 332) are not analysed and why the essential amino acid tryptophane is not even mentioned once.
A problem is also the inclusion of the ANN-modelling: what is gained from it is unclear? The authors fail to adequately, let alone convincingly, explain the usefulness of artificial neural network models in the context of studies like theirs. Artificial neural network approaches have been used for several decades now primarily in some medical/pharmaceutical, agricultural and Earth science related research, but to the best of this reviewer not (or then at least very rarely) in food-related, experimental studies, let alone edible insect studies. Apart from needing to explain the method and its applicability to the problem under investigation, the authors of this paper should refer to other food studies in which the usefulness of ANN (which - actually depending on the context it is used for- , is often seen as superior to the multilayer neural network MLP the authors employed) and explain to the reader what they expected to gain from it. As it seems, their experimental data are entirely sufficient and did not benefit at all from the ANN-modelling in this ms. Besides, the definitions and explanations of the differences of ANN2, ANN3. and ANN4 given in one short paragraph on page 4 are insufficient and a detailed discussion of the ANN-derived data in comparison with the experimentally-obtained data and conclusions is missing. In summary, readers not familiar with ANN-modelling and MLP will not be convinced that the use of these methods has much to offer in terms of improving conciseness and accuracy of the experimentally-derived results. In fact, the inclusion of the ANN/MLP material gives the impression as if the authors were ‘padding’ their manuscript. My recommendation: throw out the ANN/MLP part; its inclusion does nothing to improve your paper.
Title: the part “…as alternative feeds in contemporary animal feed production “ sounds odd. Better write “ Comparison of nutritional profiles of super worm (Zophobas morio) and yellow mealworm (Tenebrio molitor) as alternative feeds used in animal husbandry” No ‘full stop. (.) is needed after the title! The second short sentence “Is superworm superior” should not be in Capital letters and should be dropped (or added to “…animal husbandry: is superworm superior?”
Simple Summary: “Currently the food industry is facing…increase of the global human population, resulting in an increase in the demand of livestock.”
L 15: “commonly” (instead of mostly)
L22: do you mean “extended mass production” There is no such a thing as ‘ extender mass production’. Correct throughout the ms.
L23: do not write ‘larvae mill’; use “larval meal” throughout the ms.
L25: “… nutritional source and a promising alternative to traditional feed ingredients. However, optimization of the rearing….wider use in the animal feed industry.
L28: “…of the two species’ larvae and to identify ….larval meal.
L30: “…(ZM) were reared on three….after three time periods.
L33: “… optimization of the larval…. L34: …larval meal……on a mixture of…
L36-38:”…with a favourable …fatty acids.” “…in the larval diet…in an increase in all samples.”
Introduction: L43/4: “…increase in the global..”
L45: delete ‘growth’ after population
L50: Instead of ‘Nowadays’ you had better write “Ever since 1975 when Meyer-Rochow suggested that the use of edible insects as food and feed could ease the problem of global food shortages, insects have become recognized as an attractive nutritional source of food and in connection with animal husbandry feed as well. This is mainly due to the insects’ fast growth in small enclosures, high feed conversion….ammonia [ ]”
L54 / 55: what is “they” referring to? “…making its production..” what is ‘its’ referring to?
L60: you mean “expanded mass production”
L62: not ‘who’, but “that claim to have the capacity…”
L71: why do you enter here after Tenebrio molitor “( Coleoptera: Tenebrionidae)”? You mentioned the species for the first time on L56 !
L72: “…most examined insects in…” you should refer to Ghosh et al. 2016 here !
L73/74: “..are that the female lays an abundant…
L75: you write “Many studies….” But are citing only one (L78)! That simply does not tally.
L79: “…more recently investigated insects with great nutritional potential…”
L80: why here “(Coleoptera: Tenebrionidae)” since you mentioned the species already a few times earlier?
L82/83: “…but reliable scientific data regarding its mass production are still largely lacking.”
L85: You write “ In some studies…” but cite only one ! (L 87)
L95: “…that the diet has…on the nutritional value of the various insect meals [ ]” Here you would have to refer to Meyer-Rochow, V.B.; Gahukar, R.T.; Ghosh, S.; Jung, C.; Chemical Composition, Nutrient Quality and Acceptability of Edible Insects Are Affected by Species, Developmental Stage, Gender, Diet, and Processing Method. Foods 2021,10, 1036. https://doi.org/10.3390/foods10051036
L101: larval diet
Materials and Methods, L111: “…reared only on a plant-based…”
L112: “…in an insect’s diet, which …”
L120: you write ‘were placed’, but ‘placed’ where ? You mean “…were allowed to deposit eggs for three weeks.”
L121: “…identical amounts of larvae (42.60 g)..”L125: fresh ones
L126: The ‘temperature’? Where does that refer to? Temperature in the box, in the room?
L128: you did NOT have any bugs! Bugs are Hemiptera! You had “beetles (Coleoptera!)
L130: “…the larvae were inactivated…”
L132/3: The inactivating method … on the basis of a recommendation by…”
L140: “However, a kp factor…”
L151: Mineral analysis
L157: “Larval oil…with a 2:1…”
L163/4: “…FAMEs is the comparison between retention times and reference standards…”
ANN modelling: delete
MOO (lines 199-210: delete
Results, L: 213: “The nutritive profiles of….are presented…”
L215: “…when TM larvae were harvested…”
L216: you cannot write ‘which diet’! You mean “ZM, the diet of which included…”
L219/20: “…104 days, a negative effect on the CF content was discovered” (or better “…was apparent.”
L220 etc: “…TM was higher than that of ZM an substrate K and a harvest time of 97 days negatively influenced crude ash content in TM.”
L223: “…a positive …in both kinds of larvae, in which…”
L226: “…a positive effect on….for ZM, the diet of which included….” (or “…ZM, whose diet included…”
L 237: delete ‘one’
L238, etc: “…larval meals. As is shown in Fig. 2, the larval diet that included substrate S positively affected Thr, Met….TM, but negatively Val, Leu, Isl and His content in TM. In ZM a positive effect was noticed in connection with Leu, His …..ZM, following the use of substrate K in the larval diet during the 97 days of rearing”.
L243: “…hand, a negative ….was apparent in ZM…”
L245: “…K in the TM diet…”
Figures 2 2, 3, 4: the singular of species is “species” (NOT ‘specie’ !) Also: you need to write “…of the insect species’ (T. molitor and Z. morio)…” Do NOT write ‘insect’s specie’: That is wrong!
L252: “Compared with TM, ZM contained a higher amount of Ca, but had lower amounts of K, Mg…”
L259: “A negative influence on…”
L261: “…a higher level…”
L262: “A positive effect…in TM, whose diet…” (or “…, the diet of which included”)
L264/5: “…addition, a diet…had a positive…contents in TM…”
L272: “The FA profiles…are shown in Table S4. Regardless of the insect…”
L274: “…(MUFA) than TM. SFA…”
279/80: “…was highest when the larvae had consumed…”
L292 – 319: delete, it does not add anything important or of value to the results.
Discussion, L322: “In this study, TM exhibited a higher yield than ZM and it increased..” What do you mean with ‘higher yield’ ? Higher yield of what? And what increased with time of rearing? Please clarify.
L327: “…an intensive growth phase from…”
L332: “…improves an insect’s growth…”
L333: “…, it is poor in macro-nutrients…” (what is poor?)
L334: in scientific texts you never write ‘doesn’t’ or isn’t or don’t. Use “does not influence the protein content of the larvae [ ].”
L337: “..the larval meal.”
L338: “…have a higher CP…than CF due to the higher…”
L339: larval oil
L340: “In comparison with commonly used…” (Note: you use compared with or in comparison with, when you are referring to a true and real comparison. You use compared to or in comparison to, when you seek similarity. For example: the climate of Serbia can be compared to that of Korea (meaning is similar). BUT the climate of Serbia can be compared with that of Korea (a real comparison, e.g., temperatures, weather patterns sunshine hours etc are assessed).
L341: “…fishmeal, the investigated larval meal…”
L343: similar to the observed larval meals…”
L344”…accounts for more …”
L345: “..high amounts of fat and although mostly…”
L350: delete ‘with’ after ‘were fed’ and delete the ‘e’ in larvae and write “larval meals”
L354/5 “…but reported a signficant influence of the dietary fat…”
L355: you rite “Some studies…” but only cite one study on L 357.
L356 “…that the basic chemical…”
L358: did not
L359: delete ‘had’ before affected
L363: as before, here you need to cite L95 !
L370: “…lower than that of…”
L371/2: rite “…conditions the lowest ash content was obtained in ZM, which is…”
L375: “A higher amount…has a favourable influence…”
L378: Note this important publication Kipkoech e al. In Vitro Study of Cricket Chitosan’s Potential as a Prebiotic and a Promoter of Probiotic Microorganisms to Control Pathogenic Bacteria in the Human Gut. Foods 2021, 10, 2310. https://doi.org/10.3390/foods10102310
L384: delete ‘those of’ and write “…similar to that of…and a bit lower than that of fishmeal…”
L393: ”…by their higher amounts and…”
L397: “…are in agreement with our results…”
L399 etc: “…AA profiles while similar to AA profiles … meals [ ]. In comparison with fishmeal, ZM as well as TM larvae contain lower amounts of Met….. foodstuff this could be a limiting factor…”
L402/3: “genetic methods”? Which? How? What needs to be done: elaborate.
L403: larval meal
L409: compared with that of TM
L410: contents of…
L413/4: contents of …. powdered larval meal
L421/2: “…an animal’s requirements of Ca could be the utilization of a Ca-fortified diet in the insect diet.”
L423: “…CaCO3 in the insect diet can increase the Ca-content…”
L426/7: “ZM had a higher content of SA but a lower content of MUFA than TM. C18:2n6 was the dominant…” ( what does ‘considerably distributed’ mean? Do you mean was also present in ‘considerable amounts’? or ’in not exactly negligible amounts’)
L430: ‘A lot of’: can you be more precise (or less ‘colloquial’?)
L435 etc: “A more favourable level…reported ones [ , ] and it varied… (WHAT does ‘it’ refer to ? This study??)
L437: “…indicated that a diet….can enrich the C18:3n3 content in larval meals…” Instead of ‘enrich’ do you not mean ‘augment’
L446: larval meals
L455: “Comparisons between the nutritional properties and chemical compositions of T. molitor and Z. morio have shown that the latter is potentially more suitable for practical use in the feed formulation than the former”.
References: Remove unnecessary and outdated ones and update. Add those mentioned above, e.g., Meyer-Rochow, V.B. Can insects help to ease the problem of world food shortage? Search 1975, 6, 261–262.
Ghosh S, Lee S-M., Jung C., Meyer-Rochow V.B. 2017. Nutritional composition of five commercial edible insects in South Korea. Journal of Asia-Pacific Entomology 20 (2017) 686–694
Author Response
Please see the attachment (Reviewer 2)
Reviewer 3 Report
Dear Authors,
Thank you for your answers to my comments and I hope my comments will help to improve the quality of the manuscript.
However, I suggest the authors to avoid overestimation of protein content in Zophobas morio (Super worm), therefore I propose the use of 4.76 nitrogen to the protein conversion factor for the quantification of protein content in whole larvae.
Author Response
Please see the attachment (Reviewer 3)

Round 2
Reviewer 2 Report
You seem to have missed entering into the reference to the citaionon of a paper you cite n line 48: Ever since 1975 when Meyer-Rochow suggested that use of edible insects as food and feed could ease the problem of global food shortages...". Please correct and the paper is acceptable. A really important, thorough and interesting study.,